# Caspase-1 activates gasdermin A in non-mammals

Zachary Paul Billman[1,2†], Stephen Bela Kovacs[1,2†], Bo Wei[1], Kidong Kang[1], Ousmane H Cissé[3*‡], Edward A Miao[1*‡]

[1]Department of Integrative Immunobiology; Molecular Genetics and Microbiology; Pathology; and Cell Biology, Duke University School of Medicine, Durham, United States; [2]Department of Microbiology and Immunology, University of North Carolina at Chapel Hill, Chapel Hill, United States; [3]Critical Care Medicine Department, National Institutes of Health Clinical Center, Bethesda, United States

**\*For correspondence:**
ousmane.cisse@nih.gov (OHC);
edward.miao@duke.edu (EAM)

[†]These authors contributed equally to this work
[‡]These authors also contributed equally to this work

**Competing interest:** The authors declare that no competing interests exist.

**Abstract** Gasdermins oligomerize to form pores in the cell membrane, causing regulated lytic cell death called pyroptosis. Mammals encode five gasdermins that can trigger pyroptosis: GSDMA, B, C, D, and E. Caspase and granzyme proteases cleave the linker regions of and activate GSDMB, C, D, and E, but no endogenous activation pathways are yet known for GSDMA. Here, we perform a comprehensive evolutionary analysis of the gasdermin family. A gene duplication of GSDMA in the common ancestor of caecilian amphibians, reptiles, and birds gave rise to GSDMA–D in mammals. Uniquely in our tree, amphibian, reptile, and bird GSDMA group in a separate clade than mammal GSDMA. Remarkably, GSDMA in numerous bird species contain caspase-1 cleavage sites like YVAD or FASD in the linker. We show that GSDMA from birds, amphibians, and reptiles are all cleaved by caspase-1. Thus, GSDMA was originally cleaved by the host-encoded protease caspase-1. In mammals the caspase-1 cleavage site in GSDMA is disrupted; instead, a new protein, GSDMD, is the target of caspase-1. Mammal caspase-1 uses exosite interactions with the GSDMD C-terminal domain to confer the specificity of this interaction, whereas we show that bird caspase-1 uses a stereotypical tetrapeptide sequence to confer specificity for bird GSDMA. Our results reveal an evolutionarily stable association between caspase-1 and the gasdermin family, albeit a shifting one. Caspase-1 repeatedly changes its target gasdermin over evolutionary time at speciation junctures, initially cleaving GSDME in fish, then GSDMA in amphibians/reptiles/birds, and finally GSDMD in mammals.

## eLife assessment

This study presents **valuable** insights into the evolution of the gasdermin family, making a strong case that a GSDMA-like gasdermin activated by caspase-1 cleavage was already present in early land vertebrates. **Convincing** biochemical evidence is provided that extant avian, reptilian, and amphibian GSDMA proteins can still be activated by caspase-1 and upon cleavage induce pyroptosis-like cell death -- at least that they do so in the context of human cell lines. The caspase-1 cleavage site has only been lost in mammals, which use the more recently evolved GSDMD as a caspase-1 cleavable pyroptosis inducer. The presented work will be of considerable interest to scientists working on the evolution of cell death pathways, or on cell death regulation in non-mammalian vertebrates.

## Introduction

Gasdermin family proteins serve a critical role in innate immune defense against intracellular infection. When these proteins are cleaved within the linker region, the N-terminal domain is freed from the autoinhibitory C-terminal domain (*Kovacs and Miao, 2017*). The liberated N-terminus polymerizes into a ring that inserts into the plasma membrane, forming a pore that causes lytic cell death called pyroptosis. This eliminates infected host cells to clear intracellular infections. The best studied gasdermin is gasdermin D (GSDMD) which is activated by caspase-1 in response to diverse stimuli (*Devant and Kagan, 2023*).

During infection, pathogens manipulate the host cell, sometimes contaminating the cytosol with pathogen-associated molecular patterns, or disrupting aspects of normal cell physiology. These signs can be detected by cytosolic sensors. These sensors oligomerize to form a platform that activates the protease caspase-1 (*Nozaki et al., 2022a*). In parallel, caspases-4/5 (human) or -11 (mouse) directly detect and activate in response to cytosolic lipopolysaccharide as a proxy for microbial contamination. These activated caspases then cleave GSDMD to trigger pyroptosis (*Shi et al., 2015*; *Kayagaki et al., 2015*). This activation of gasdermin D by caspase-1/4/5/11 to engage pyroptosis is well conserved in mammals and is essential for defense against many infections, including environmental pathogens that would otherwise be deadly (*Li et al., 2023*).

At its discovery, pyroptosis was defined as lytic regulated cell death caused by caspase-1 (*Cookson and Brennan, 2001*), and later caspases-4/5/11. This original definition was expanded upon the discovery of the gasdermin family to include any gasdermin-driven lytic cell death regardless of the initiating pathway (*Shi et al., 2015*). Indeed, mammals encode *GSDMA*, *GSDMB*, *GSDMC*, *GSDMD*, and *GSDME*, which have all been demonstrated to form pores to trigger pyroptosis (*Kovacs and Miao, 2017*). A related protein, pejvakin (PJVK), has a similar N-terminal domain but a smaller C-terminal domain, and has not yet been demonstrated to form pores.

Since the discovery of gasdermins as pore-forming proteins (*Ding et al., 2016*), many have sought to identify the activation mechanisms of these proteins. GSDME was next found to be activated by caspase-3 (*Wang et al., 2017*; *Rogers et al., 2017*). GSDMB was shown to be cleaved and activated by granzyme A after cytotoxic lymphocyte attack (*Zhou et al., 2020*), offloading the death decision to killer cells and activating GSDMB in a cell extrinsic manner. GSDMC was later shown to be activated by caspase-8 (*Hou et al., 2020*; *Zhang et al., 2021*), allowing a cell to attach TNF signaling to pyroptosis. GSDMA remains outlier, because the only known activator is the group A Streptococci secreted protease SpeB rather than a host protease (*LaRock et al., 2022*; *Deng et al., 2022*).

Only mammals encode *GSDMC* and *GSDMD* alongside the other four gasdermins. Other clades of animals encode fewer gasdermin genes (*Yuan et al., 2022*; *De Schutter et al., 2021*; *Wang et al., 2023b*; *Angosto-Bazarra et al., 2022*). The clades that encode the fewest gasdermins that includes *GSDMA* are birds and amphibians, which express just *GSDMA*, *GSDME*, and *PJVK*. Understanding how these organisms utilize fewer gasdermins, especially the newly emerged *GSDMA*, provides an opportunity to better understand this important gene family.

## Results

### Evolutionary analysis reveals the origins of mammal gasdermins

To understand the evolution of the gasdermin family we reconstructed a phylogeny of gasdermin proteins. We screened a total of 3443 genomes covering major classes of animal orders from public databases using gasdermin-specific hidden Markov models to identify all putative gasdermin genetic loci using trained gene predictors. We uncovered 106 previously unannotated GSDM proteins. Our final catalog included 1256 gasdermin protein sequences. Gasdermin-like proteins have been reported in bacteria in phage defense islands alongside activating proteases (*Johnson et al., 2022*; *Wein and Sorek, 2022*), and in fungi (*Daskalov et al., 2020*). We limited our searches to animals because sequences from other kingdoms are distantly similar (*Yuan et al., 2022*; *Angosto-Bazarra et al., 2022*). Our approach extends previous studies by using a more comprehensive dataset, which allows us to more accurately retrace gasdermin gene family evolution withing the context of vertebrate radiation.

In comparison to previously published studies, we used different methods to construct our gasdermin phylogenetic tree, with the result that our tree has a different topology. The topology of

our tree is likely to be affected by our increased sampling of gasdermin sequences; we included 1256 gasdermin sequences in comparison to 300 or 97 sequences used in prior studies. Prior studies used maximum likelihood tree building techniques, whereas we used a more comprehensive estimation of gasdermin divergence time using Bayesian method implemented in BEAST with strict molecular clocks using fossil calibrations with soft bounds. We think that this substantially increased sampling paired with time calibration allow us to produce a more accurate phylogeny of the gasdermin protein family. We constructed a well-supported Bayesian tree with 71% of the nodes with posterior probability >0.8 and a minimum age of protogasdermin emergence at 1100 million years with 95% higher-posterior density intervals (*Figure 1*).

Similar to other trees, GSDME and PJVK cluster together in a clade separate from the clade containing GSDMA–D and GSDMEc. However, the internal order within this second clade was divergent from other trees, described in more detail below.

Near the midpoint of our tree is a gasdermin that is preserved in mollusks, coral, hydras, and other invertebrates referred to as $GSDM_{in}$. For such species, caspase-3 cleaves this gasdermin to execute pyroptosis, much like GSDME (*Jiang et al., 2020*; *Qin et al., 2023*; *Chen et al., 2023*).

The first gene duplication of this gasdermin occurred after the divergence of lancelets and chordates resulting in GSDME and PJVK. Bony fish encode both caspase-1 and -3, and both cleave GSDME to execute pyroptosis (*Jiang et al., 2019*; *Chen et al., 2021b*; *Xu et al., 2022*; *Chen et al., 2021a*; *Wang et al., 2020a*; *Li et al., 2020*). Similarly, lancelets cleave GSDME using caspase-1- and caspase-3-like (*Wang et al., 2023b*). In contrast, PJVK developed a new function of promoting peroxisome proliferation in response to excess reactive oxygen species, utilizing a shorter C-terminus that detects ROS (*Delmaghani et al., 2015*; *Defourny et al., 2019*). Although it has not been shown yet, this function is likely related to a pore-forming ability. We also found additional ancient gasdermin sequences in bony fish with a GSDM C-terminal domain, but sometimes without the GSDM N-terminal domain, that appeared in a clade that branches off before the divergence of GSDME and PJVK (*Figure 1*, purple, GSDMEc-like).

An ancient ancestor of *GSDME* was duplicated and gave rise to the second gasdermin clade (sometimes called the GSDMA family). This gasdermin is not universally present in bony fish, but in some it is preserved as *GSDMEc* (*Yuan et al., 2022*). This gasdermin was recently identified as a potential ortholog of GSDMA. It was called GSDMEc, following the nomenclature of other duplications of GSDME in bony fish which have been named GSDMEa and GSDMEb. Our method brought to light this gasdermin is present broadly in sharks, which appeared in a separate clade near the root of the GSDMA family proteins (*Figure 1*, purple, GSDMEc Shark). This discovery places the emergence of *GSDMEc* at the common ancestor of jawed vertebrates. The activating protease for the GSDMEc proteins remains undetermined, but they do not contain apparent caspase cleavage motifs.

*GSDMEc* then gave rise to *GSDMA* in the common ancestor of amphibians, mammals, reptiles, and birds (*De Schutter et al., 2021*; *Angosto-Bazarra et al., 2022*; *Yuan et al., 2022*). No additional gasdermins are found in amphibians or birds, therefore, these clades encode *GSDME*, *PJVK*, and *GSDMA*. Among these, GSDME is activated by caspase-3 to execute pyroptosis (*Li et al., 2022*). Our and other molecular phylogeny analysis identified three amphibians in the caecilian order with *GSDMA* genes (*Yuan et al., 2022*; *Wang et al., 2023b*). *GSDMA* is absent in frogs and salamanders, including amphibian model organisms with well characterized genomes like *Xenopus tropicalis* (amphibian, Western clawed frog). This observation suggests that *GSMDA* is present in the common ancestor of tetrapods, but was lost in most amphibians. In support of this, caecilian *GSDMA* resides in the same locus as mammal, reptile, and bird *GSDMA*. The proteases that activate GSDMA in these clades had not been investigated.

We and others identified a poorly characterized gasdermin in the *GSDMB* locus, which is separated by only two genes from *GSDMA* which we have labeled as GSDMB-like, despite little sequence similarity with mammal GSDMB (*Wang et al., 2023b*). This gasdermin grouped in a clade alongside fish and shark GSDMEc. The presence of this gasdermin in the same locus as mammal *GSDMB* in reptiles before and after the divergence of turtles, crocodiles, and birds from lizards and snakes suggests that this gasdermin was present in the common ancestor reptiles and birds.

Previous trees suggest that mammal GSDMA and GSDMB evolved in a separate clade from GSDMC and GSDMD (*De Schutter et al., 2021*; *Wang et al., 2023b*). Surprisingly, in our tree, amphibian, bird, and reptile GSDMA are paralogs to mammal GSDMA rather than orthologs. Both our Bayesian

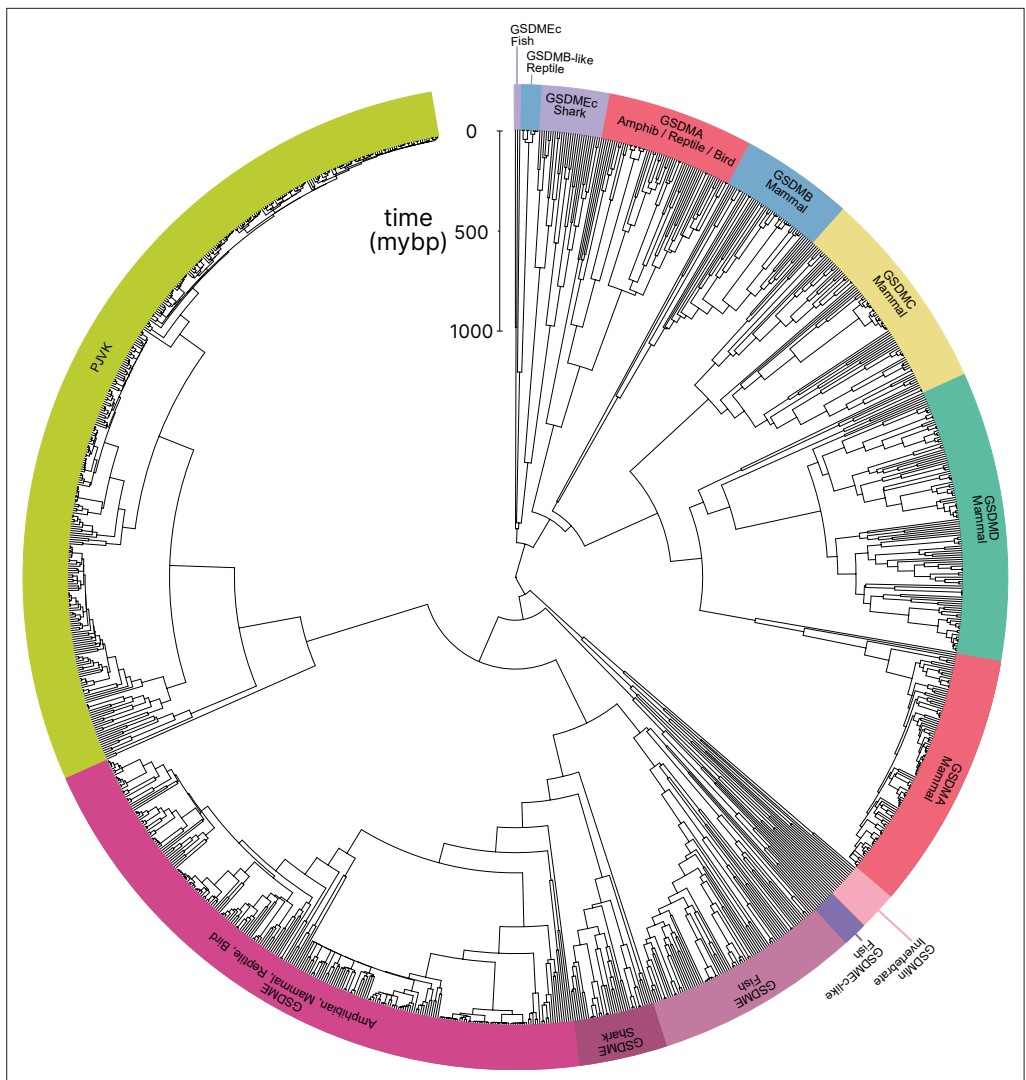

**Figure 1.** Phylogenetic analysis of GSDM proteins. Time-calibrated maximum clade credibility coalescent Bayesian phylogenetic tree of all protein sequences in selected superphyla by hidden Markov model identification method. Tree was visualized using iTOL v6 (***Letunic and Bork, 2021***).

The online version of this article includes the following source data and figure supplement(s) for figure 1:

**Source data 1.** Tree in .tre format.

**Source data 2.** Alignment file of all sequences in nexus format.

**Figure supplement 1.** Tree identical to ***Figure 1*** with individual nodes labeled.

**Figure supplement 1—source data 1.** Text file with all sequences with seqID and ID used in nexus alignment file.

**Figure supplement 2.** Tree identical to ***Figure 1*** with individual nodes labeled and posterior probabilities displayed at the midpoint of each node.

**Figure supplement 3.** Maximum likelihood tree generated using the same sequences in ***Figure 1*** Bayesian tree.

**Figure supplement 3—source data 1.** Maximum likelihood tree in Newick format.

**Figure supplement 3—source data 2.** Maximum likelihood tree with bootstrap values.

**Figure supplement 3—source data 3.** Maximum likelihood tree with node tips labeled.

**Figure supplement 3—source data 4.** Maximum likelihood tree with node tips labeled and bootstrap values.

**Figure supplement 4.** Visualization of the locus where *GSDMA* is found in the clades of animals that encode this gasdermin.

(*Figure 1*) and maximum likelihood (*Figure 1—figure supplement 3*) trees produced this result, suggesting this placement is due to increased sampling of gasdermin sequences. Importantly, the *GSDMA* gene is found in the same locus in all animals in these clades (*Figure 1—figure supplement 4*). This suggests that the common ancestor of amphibians and mammals duplicated *GSDMA* first to give rise to mammal *GSDMB*, then sequentially gave rise to *GSDMC*, and then *GSDMD*, leaving the syntenic *GSDMA* gene at the ancestral locus.

The absence of *GSDMB* and *GSDMC* in the ancient monotreme mammals at first glance appears to be in conflict with this sequence of evolution. Our tree reconciles this observation by suggesting that these gasdermins were lost in the monotreme lineage represented by *Ornithorhynchus anatinus* (monotreme mammal, platypus) in our tree.

Marsupials and placental mammals can encode all of *GSDME, PJVK, GSDMA, GSDMB, GSDMC,* and *GSDMD*. Mammals conserved the typical caspase-3 activation of GSDME (*Wang et al., 2017*; *Rogers et al., 2017*; *Van Rossom et al., 2015*). In mammals, GSDMA is cleaved by SpeB (*Deng et al., 2022*; *LaRock et al., 2022*). GSDMB is activated by granzyme A delivered by cytotoxic lymphocytes (*Wang et al., 2023a*; *Zhong et al., 2023*; *Yin et al., 2023*). GSDMC is activated by caspase-8 (*Hou et al., 2020*). Lastly, GSDMD is activated primarily by caspase-1 and related proteases (*Kayagaki et al., 2015*; *Ding et al., 2016*; *He et al., 2015*; *Kambara et al., 2018*; *Burgener et al., 2019*; *Sarhan et al., 2018*; *Orning et al., 2018*).

Numerous mammals have lost *GSDMB* and *GSDMC*. Previous studies reported this occurred in the monophyletic group that comprises mice, rats and rabbits. We identified the remnant *GSDMB* in rabbits which is interrupted by an ORM-1 like protein, as well as *GSDMB* in hamsters. This definitively shows the existence of *GSDMB* in this clade and suggests that there are multiple instances of its loss. This captures a snapshot of the rapid evolution of GSDM family proteins in mammals.

Overall, we discovered that the *GSDMA* common ancestor of amphibians, birds, and reptiles gave rise to mammal *GSDMA-D*. This is surprising given that GSDMA from both of these clades appear in the same locus (*Figure 1—figure supplement 3*). We speculated that this divergence in grouping reflected a divergence in function.

## Bird GSDMA linker contains a likely caspase-1 cleavage site

Unlike other trees (*Yuan et al., 2022*; *De Schutter et al., 2021*; *Angosto-Bazarra et al., 2022*; *Wang et al., 2023b*), we found that amphibian, bird, and reptile GSDMA are paralogs to mammal GSDMA, not orthologs, meaning we found that they appear in separate clades. Using the wealth of gasdermin sequences we assembled for our phylogenetic analysis, a remarkable trend in the bird GSDMA sequences became clear: nearly a third of all birds encoded a YVAD-like tetrapeptide in the linker region (*Figure 2A, B*). This sequence is famously cleaved by caspase-1 in mammals and is the peptide sequence utilized as an inhibitor and fluorometric substrate for measuring caspase-1 activity (*Garcia-Calvo et al., 1998*). The remaining bird GSDMA sequences contained a similar tetrapeptide also predicted to be cleaved by caspase-1, with a bulky P4 residue and aspartic acid at P1, most commonly FVSD or FASD (*Figure 2B*; *Agard et al., 2010*; *Shen et al., 2010*; *Timmer and Salvesen, 2007*; *Thornberry et al., 1997*). Amphibians, lizards, and snakes have similar tetrapeptides, albeit with more variation (*Figure 2B*).

An alignment of the linker regions of GSDMA proteins from selected birds, reptiles, and amphibians revealed conservation of these tetrapeptide sequences, despite differences in the length and overall lack of conservation within the linker (*Figure 2C*). These tetrapeptides appear similar to those commonly seen in the mammal GSDMD sequences, for example FLTD (human) and LLSD (mouse). Additionally, the P1′ residue in amphibian, bird, and reptile GSDMA was often a glycine, and the P2′ residue was often a proline, especially in birds with FASD/FVSD tetrapeptides (*Figure 2B*). A small P1′ residue is preferred by all caspases (*Stennicke et al., 2000*). By using a peptide library, glycine has been determined to be the optimal P1′ residue for caspase-1 and -4 (*Talanian et al., 1997*). Furthermore, in a review of the natural substrates of caspase-1, glycine was the second most common P1′ residue, and proline was the most common P2′ residue. These preferences were not observed for caspase-9 (*Shen et al., 2010*).

Amphibians, reptiles, and birds all express caspase-1 and have expanded their gasdermin family beyond GSDME to include GSDMA but notably lack GSDMD. We hypothesized that GSDMA may functionally mimic mammal GSDMD.

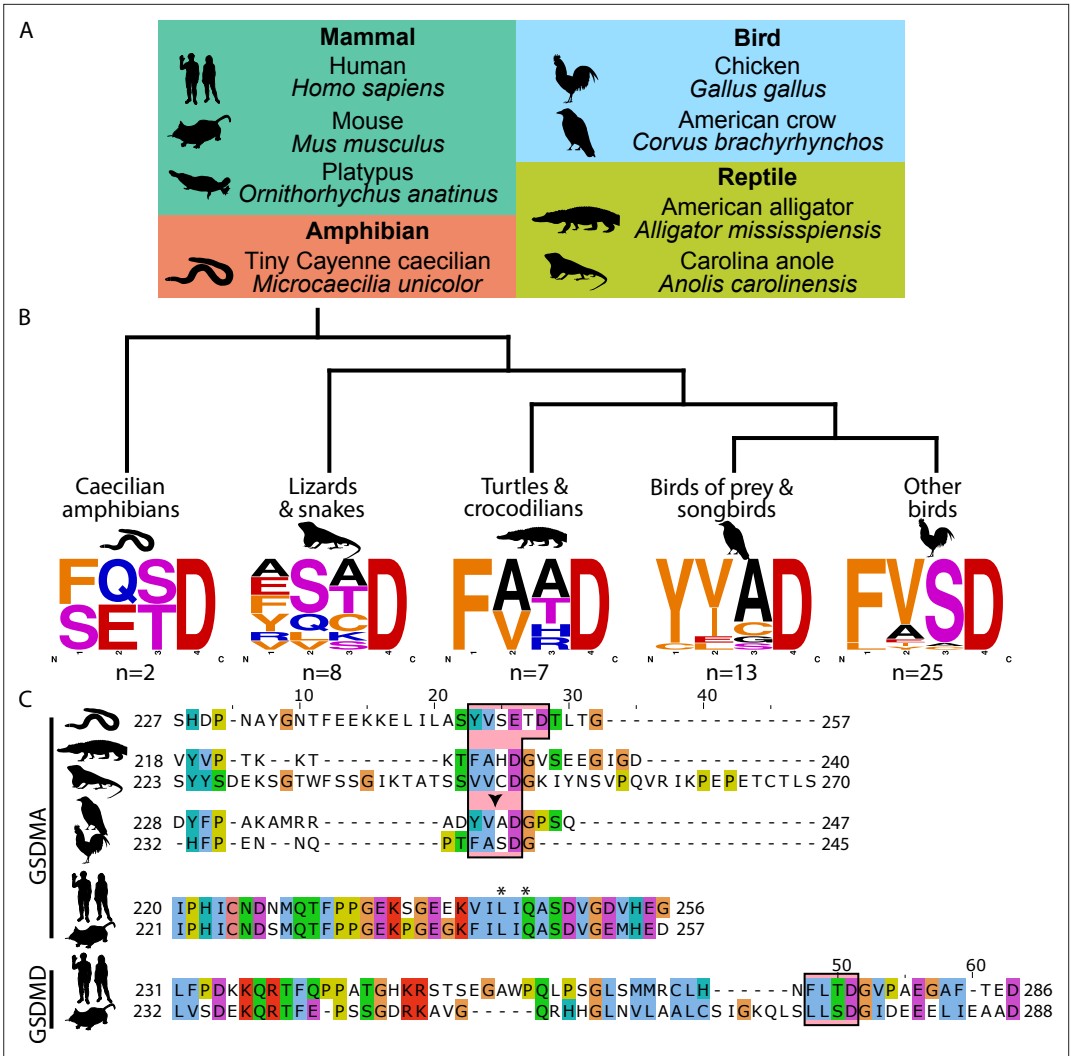

**Figure 2.** Analysis of bird GSDMA sequences suggests caspase-1 cleavage. (**A**) Cartoons used in this study to represent organisms and their phylogenetic classification. Note that caecilian amphibians morphologically are similar to snakes in their lack of legs and elongated body, however, this is an example of convergent evolution as caecilians are amphibians and are thus more closely related to frogs and salamanders than snakes. (**B**) Sequence logo plots derived from the tetrapeptides found in amphibian, bird, and reptile GSDMA organized by evolutionary clades. Importantly, 'Other birds' is not a monophyletic group. Sequence logos display abundance of amino acids in each position before an aspartic acid in the linker region of each GSDMA protein using WebLogo (*Crooks et al., 2004*). When more than one aspartic acids were present, multiple tetrapeptides were listed. (**C**) Alignment of selected amphibian, reptile, bird, and mammal GSDM linker regions by MUSCLE. Linker defined as region between β11 and α5 by SWISS-MODEL. Outlined in a red box are tetrapeptide sequences that precede caspase-1 cleavage sites for mammal, or show similarity in amphibian, reptile, and bird sequences. The canonical caspase-1 cleavage site YVAD is emphasized with an arrowhead. '*' denotes cleavage sites of SpeB for human and mouse GSDMA. Amino acids colorized using Clustal color scheme.

The online version of this article includes the following source data for figure 2:

**Source data 1.** FASTA file with sequences of all GSDMAs in this figure.

**Source data 2.** FASTA file with sequences of tetrapeptides to generate sequence logos.

## Bird *GSDMA* expression mimics mammal *GSDMD* expression

We focused our studies on birds, where the abundance of available sequences is high, and tissues are readily available. We first evaluated the expression of *GSDMA* and *GSDME* in *Gallus gallus* (chicken, which encodes an FASD tetrapeptide) because it is one of the best studied bird species and tissues are readily available for study. The gasdermin family proteins were so named because of their expression

in the gastrointestinal tract (gas-) and the skin (-derm), with *GSDMA* being specifically expressed in the skin and upper gastrointestinal tract in mice (*Tamura et al., 2007*). If bird GSDMA functions similar to mammal GSDMD, then we would expect their expression profiles to be similar. Mammal *GSDMD* is primarily expressed broadly in immune cells as well as epithelial cells (*De Schutter et al., 2021*). Quantitative reverse transcription PCR (qRT-PCR) analysis from various chicken tissues revealed that chicken *GSDMA* was preferentially expressed in the Bursa of Fabricius (bird-specific primary lymphoid organ) and the spleen (*Figure 3A*). These data suggest that *GSDMA* is preferentially expressed in immune cells, similar to the expression of *GSDMD* in mammals. We additionally evaluated these same tissue sites for expression of *CASP1* and *GSDME* and found that they were expressed similarly across most tissues (*Figure 3—figure supplement 1*). Interestingly, unlike mammal *GSDMA*, chicken *GSDMA* was not enriched in the skin. These expression data, together with the caspase-1 cleavage site in the linker region suggest that bird GSDMA may serve as their GSDMD.

## Bird GSDMA is cleaved by caspase-1

In the absence of *GSDMD*, we hypothesized that caspase-1 may recognize and cleave the linker region of GSDMA in amphibians, reptiles, and birds, the gasdermin that gave rise to GSDMD. We next tested whether caspase-1 can cleave bird GSDMA. Caspase-1 is activated upon homodimerization through its CARD domain (*Thornberry et al., 1992*). To simulate this in vitro, we replaced the chicken caspase-1 CARD domain with 2xDmrB domains that can be homodimerized by the addition of the small molecule AP20187 (*Figure 3B*; *Burnett et al., 2004*; *Freeman et al., 2003*). We transfected 293T/17 cells with chicken 2xDmrB-caspase-1 and 3xFLAG-GSDMA (contained an FASD tetrapeptide). When we treated these cells with the AP20187 dimerizer, we observed strong intensification of the N-terminal GSDMA band (*Figure 3C*) that increased in a time-dependent manner (*Figure 3D*). The weak cleavage observed in the absence of dimerizer likely resulted from overexpression-dependent autoactivation. Importantly, mutating the predicted cleavage site residue of chicken GSDMA, D244A, prevented caspase-1-dependent cleavage (*Figure 3E*), confirming that chicken GSDMA is cleaved at 240FASD244 by caspase-1.

We then observed that crow GSDMA (containing a YVAD tetrapeptide) was similarly cleaved by chicken caspase-1 (*Figure 3F*). This cleavage was abolished when the crow YVAD is mutated to YVAA. Thus, this cleavage appears to be conserved across bird species. Interestingly, we observed two N-terminal GSDMA fragments, both of which were ablated by the YVAA mutation (*Figure 3F*). This is similar to how cnidarian and mollusk GSDM release two N-terminal fragments when cleaved by caspase-3. Interestingly, in those species, both fragments are competent to form pores (*Jiang et al., 2020*; *Qin et al., 2023*). Alternatively, the second cleavage may be an inactivation site (*Xu et al., 2022*; *Taabazuing et al., 2017*).

## Bird GSDMA is activated by mammal caspase-1 and inactivated by caspase-3

We next tested the ability of human and mouse caspase-1 to cleave chicken GSDMA, and observed that both of these mammal caspase-1 proteins cleaved chicken GSDMA (*Figure 3F*). This cleavage was again abolished by mutation of the aspartate cleavage site (*Figure 3F*). This cleavage was also seen with purified human caspase-1 (*Figure 3—figure supplement 2*). Human and mouse caspase-1 also cleaved crow GSDMA, again dependent upon the YVAD tetrapeptide (*Figure 3E*). Chicken caspase-1 is also able to cleave both human and mouse GSDMD at the expected site (*Figure 3G, H*). Thus, caspase-1 activity well is conserved in diverse species. None of the tested caspase-1 proteases could cleave human GSDMA (*Figure 3I*).

We hypothesized that activating pyroptosis using separate gasdermins for caspase-1 and -3 is a useful adaptation and allows for fine-tuning of these separate pathways. In mammals, this separation depends on the activation of GSDMD by caspase-1 and the activation of GSDME by caspase-3. Birds, too, encode caspase-1 and -3. In species with as distant common ancestors as coral, GDSME is cleaved by caspase-3 (*Qin et al., 2023*; *Chen et al., 2023*; *Jiang et al., 2020*). In mammals, caspase-3 inactivates GSDMD by cleaving within the N-terminal pore-forming domain to prevent pore formation during apoptosis. Bird GSDMA proteins appear to have a caspase-3 tetrapeptide sequence at a similar location in the N-terminus. Indeed, when we activated human caspase-3 via caspase-9 (*Straathof et al., 2005*) we observed the inactivating cleavage event (*Figure 3J*, *Figure 3—figure*

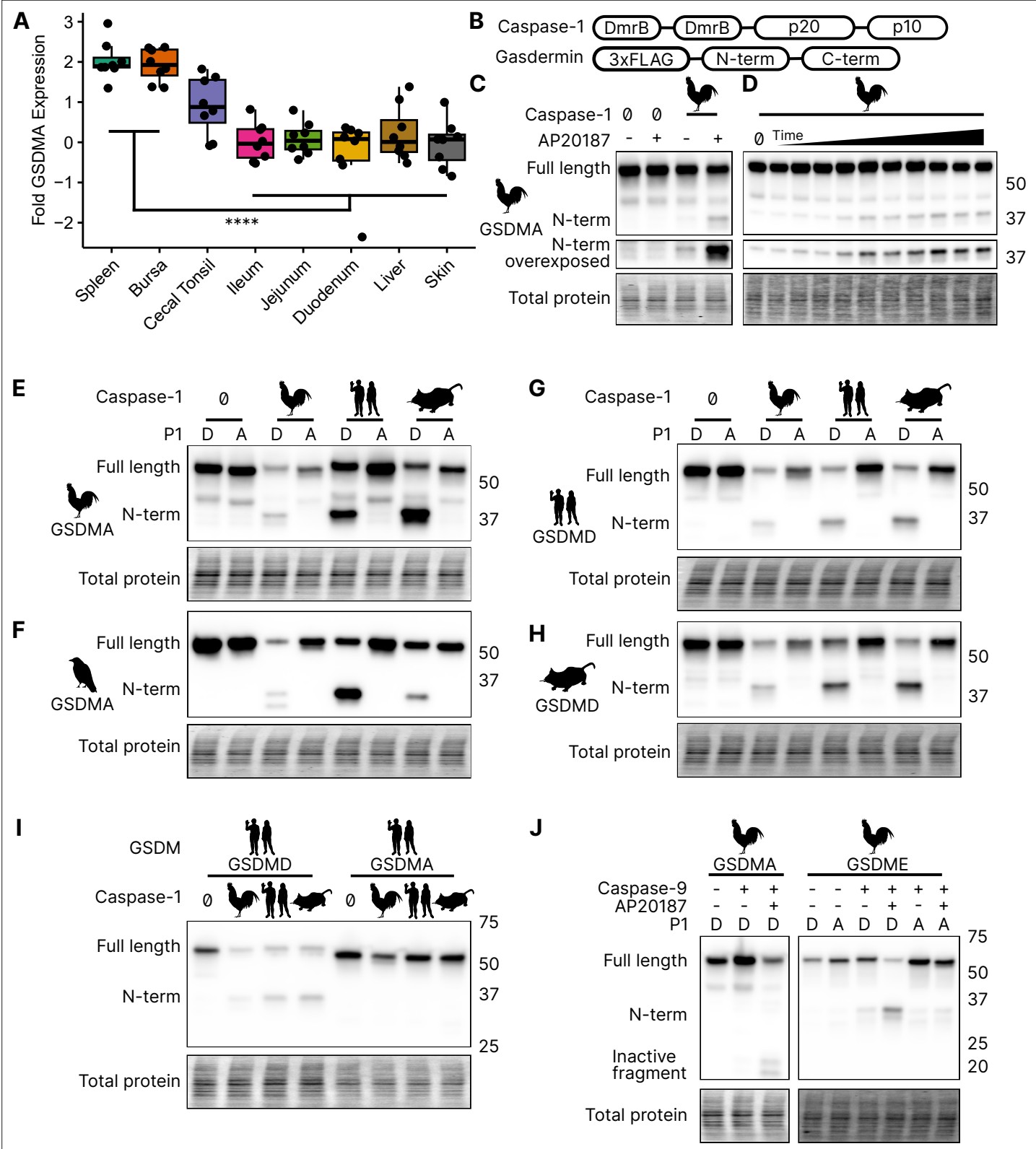

**Figure 3.** Chicken caspase-1 cleaves chicken GSDMA and mammal GSDMD. (**A**) Normalized relative *GSDMA* abundance by qRT-PCR using RNA derived from various *Gallus gallus domesticus* tissues. The cecal tonsils are a lymphoid aggregate tissue found in the chicken gastrointestinal tract, akin to Peyer's patches in mammals. ****, p < 0.0001 by two-way analysis of variance (ANOVA) with Tukey's multiple comparisons correction. Experiment was performed with technical triplicate with six biological samples. (**B**) Diagram of constructs used in this study. Note that CARD domain of caspase-1 is not

*Figure 3 continued on next page*

*Figure 3 continued*

present. (**C**) FLAG blot of HEK293T/17 lysates after co-transfection of chicken GSDMA and caspase-1 after addition of dimethyl sulfoxide (DMSO) or AP20187 dimerizer. (**D**) FLAG blot of time course of dimerizer addition. Samples were harvested at 0, 0.25, 0.5, 0.75, 1, 1.5, 2, 2.5, 3, 4.5, and 6 hr after addition of AP20187 dimerizer. (**E–H**) Caspase-1 from chicken, human, and mouse cleave bird GSMDA and mammal GSDMD. FLAG blot of HEK293T/17 lysates after co-transfection with indicated caspase-1 and GSDM constructs. P1 refers to the fourth reside of the tetrapeptide sequence, which is cleaved by caspase-1. P1 mutants are chicken D244A, crow D243A, human D275A, and mouse D276A. (**I**) FLAG blot of HEK293T/17 lysates after co-transfection with indicated gasdermin and caspase-1. Caspase-1 from chicken, human, and mouse cleave human GSDMD but not human GSMDA. (**J**) FLAG blot of HEK293T/17 lysates after co-transfection with indicated gasdermin and caspase-9. P1 mutant of chicken GSDME is $_{270}$DAVD$_{273}$ to $_{270}$DAVA$_{273}$. Blots were performed in biological triplicate.

The online version of this article includes the following source data and figure supplement(s) for figure 3:

**Source data 1.** Quantitative PCR data used to generate plot in *Figure 3A*.

**Source data 2.** Original file for the western blot analysis in *Figure 1C* (anti-FLAG).

**Source data 3.** PDF containing *Figure 1C* and original scans of the relevant western blot analysis (anti-FLAG) with highlighted bands and sample labels.

**Source data 4.** Original file for the western blot analysis in *Figure 1D* (anti-FLAG).

**Source data 5.** PDF containing *Figure 1D* and original scans of the relevant western blot analysis (anti-FLAG) with highlighted bands and sample labels.

**Source data 6.** Original file for the western blot analysis in *Figure 1E* (anti-FLAG).

**Source data 7.** PDF containing *Figure 1E* and original scans of the relevant western blot analysis (anti-FLAG) with highlighted bands and sample labels.

**Source data 8.** Original file for the western blot analysis in *Figure 1F* (anti-FLAG).

**Source data 9.** PDF containing *Figure 1F* and original scans of the relevant western blot analysis (anti-FLAG) with highlighted bands and sample labels.

**Source data 10.** Original file for the western blot analysis in *Figure 1G* (anti-FLAG).

**Source data 11.** PDF containing *Figure 1G* and original scans of the relevant western blot analysis (anti-FLAG) with highlighted bands and sample labels.

**Source data 12.** Original file for the western blot analysis in *Figure 1H* (anti-FLAG).

**Source data 13.** PDF containing *Figure 1H* and original scans of the relevant western blot analysis (anti-FLAG) with highlighted bands and sample labels.

**Source data 14.** Original file for the western blot analysis in *Figure 1I* (anti-FLAG).

**Source data 15.** PDF containing *Figure 1I* and original scans of the relevant western blot analysis (anti-FLAG) with highlighted bands and sample labels.

**Source data 16.** Original file for the western blot analysis in *Figure 1J* (GSDMA, anti-FLAG).

**Source data 17.** PDF containing *Figure 1J* and original scans of the relevant western blot analysis (GSDMA, anti-FLAG) with highlighted bands and sample labels.

**Source data 18.** Original file for the western blot analysis in *Figure 1J* (GSDME, anti-FLAG).

**Source data 19.** PDF containing *Figure 1J* and original scans of the relevant western blot analysis (GSDME, anti-FLAG) with highlighted bands and sample labels.

**Figure supplement 1.** The same tissues as in *Figure 3* were assayed for *GSDME* and *CASP1* alongside *GSDMA*.

**Figure supplement 1—source data 1.** Quantitative PCR data.

**Figure supplement 2.** Incubation of 293T/17 lysates transfected with C-terminally tagged crow GSDMA, crow GSDME, human GSDMA, or human GSDMD then were incubated with human CASP1 or CASP3.

**Figure supplement 2—source data 1.** Original file for the western blot analysis in *Figure 3—figure supplement 2* (anti-FLAG).

**Figure supplement 2—source data 2.** PDF containing *Figure 3—figure supplement 2* and original scans of the relevant western blot analysis (anti-FLAG) with highlighted bands and sample labels.

*supplement 2*), demonstrating this crosstalk exists in species that use a gasdermin besides GSDMD for caspase-1-driven pyroptosis. Also conserved in chickens is the activation of GSDME via caspase-3, which was abolished by mutating the P1 aspartic acid (*Figure 3J*, *Figure 3—figure supplement 2*).

## Bird GSDMA cleavage by caspase-1 causes pyroptosis

To confirm that the N-terminal fragment of chicken GSDMA forms a pore and causes pyroptosis, we transfected 293T/17 cells with a plasmid encoding the N-terminal domain of chicken GSDMA or the full-length protein. The N-terminal domain resulted in 85% reduction in host cell viability (*Figure 4A*). To directly demonstrate that caspase-1 cleavage of GSDMA results in cell death, we co-expressed these proteins in 293T/17 cells. Dimerization with AP20187 caused cell death that was abrogated

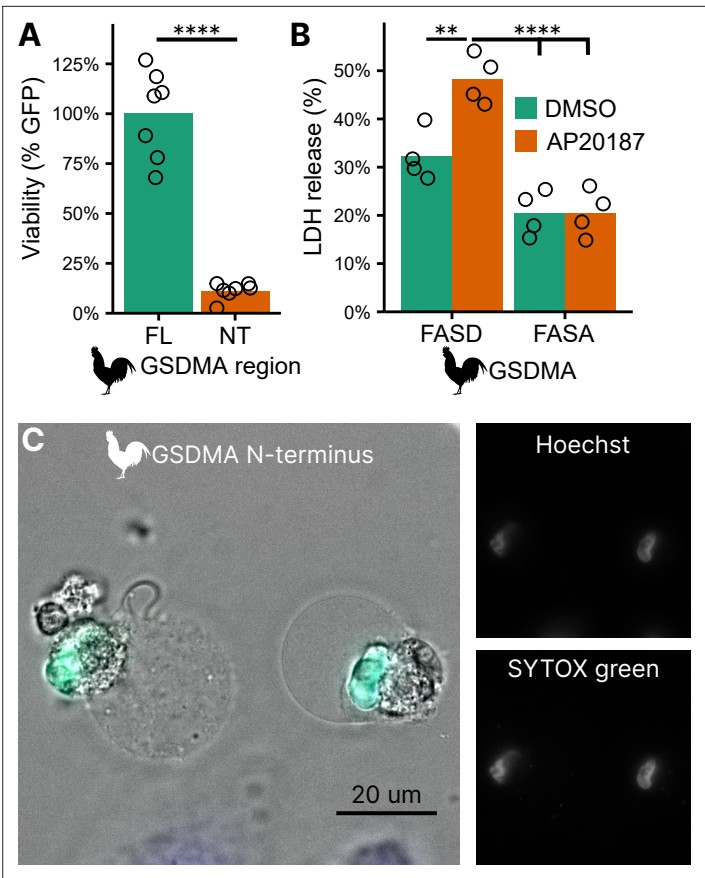

**Figure 4.** Cleavage of chicken GSDMA leads to cell lysis. (**A**) Measurement of green fluorescent protein (GFP) fluorescence in HEK293T/17 cells co-transfected with GFP and either full-length (FL) chicken GSDMA or its N-terminal (NT) pore-forming domain after 48 hr. Each point represents a separate transfection experiment performed in technical quadruplicate. ****, p < 0.0001 by Student's *t*-test. (**B**) Cell lysis measured by lactate dehydrogenase (LDH) release in HEK293T/17 cells co-transfected with chicken caspase-1 and either WT or D244A chicken GSDMA. Gasdermin constructs in this study do not encode the 3xFLAG domain. Each point represents a separate transfection experiment performed in technical quadruplicate. **, p < 0.01; ****, p < 0.0001 by two-way analysis of variance (ANOVA) with Tukey's multiple comparisons correction. (**C**) Images of HeLa cells after transfection with the N-terminal domain of chicken GSDMA. Transfected cells display classic pyroptotic ballooning. SYTOX green staining of the nucleus indicates permeabilization of cell membranes.

The online version of this article includes the following source data for figure 4:

**Source data 1.** GFP fluorescence data.

**Source data 2.** LDH release data.

when the FASD cleavage site was mutated to FASA (*Figure 4B*). HeLa cells displayed pyroptotic morphology with ballooning membrane formations when transfected with the chicken GSDMA N-terminal domain (*Figure 4C*). In these experiments, 293T/17 cells were used for their high transfection efficiency, and HeLa cells were used in microscopy studies for their larger size and improved adherence. Overall, our data suggest that the activation of chicken caspase-1 can lead to the cleavage of GSDMA to trigger pyroptosis.

## Bird GSDMA tetrapeptide sequence is essential for caspase-1 cleavage

Surprisingly counter the basic dogma of caspase enzymatic biology, it was recently discovered that caspase-1 cleaves GSDMD after the P1 aspartic acid independent of the tetrapeptide sequence (*Wang et al., 2020b*). The specificity is instead mediated by extensive interactions between an exosite in GSDMD that interfaces with caspase-1. Thus, the human GSDMD tetrapeptide FTLD can be mutated to AAAD without compromising caspase-1 cleavage (*Wang et al., 2020b*; *Liu et al., 2020*).

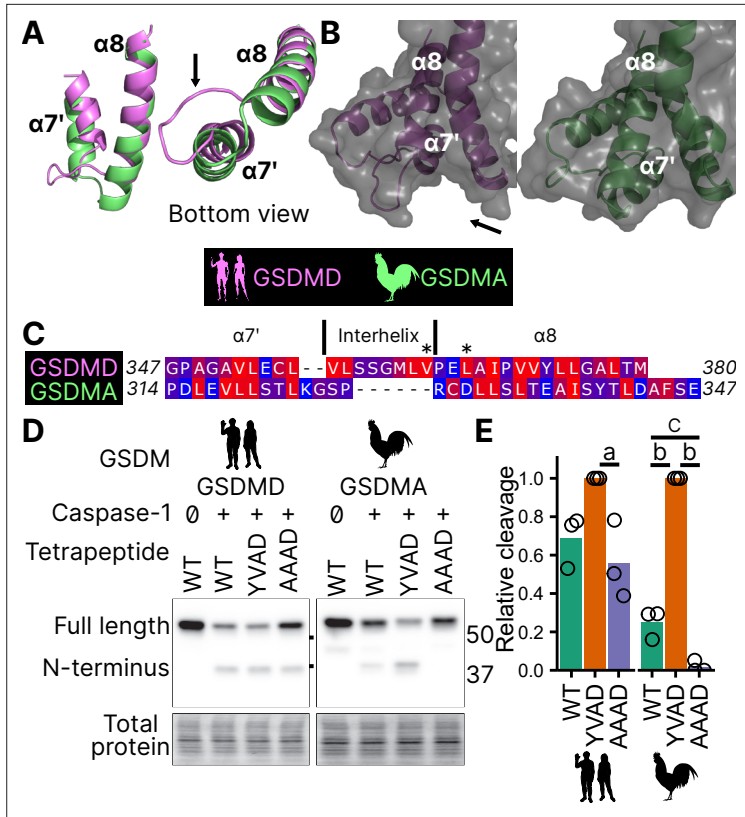

**Figure 5.** Chicken GSDMA requires a compatible tetrapeptide for activation. (**A**) AlphaFold predictions of the structure of α7' and α8 helices of human GSDMD and chicken GSMDA. α8 helices were aligned in Pymol. The human interhelix loop is indicated by the arrow. (**B**) Gasdermin C-terminus AlphaFold predictions visualized with surface projection. Structures are aligned at the α8 helix. The arrow indicates the human interhelix loop and the space where human caspase-1 interacts with the C-terminal domain of human GSDMD PDB: 6KN0 (**Wang et al., 2020b**). (**C**) Alignment of α7'-interhelix-α8 helix regions of human GSDMD (residues 347–380) and chicken GSDMA (residues 314–347). Residues colored with the Kyte and Doolittle scale with red indicating hydrophobic and blue indicating hydrophilic. '*' indicates residues determined to be critical for the hydrophobic interaction between human GSDMD and caspase-1. (**D**) FLAG blot of HEK293T/17 cell lysates after co-transfection with either chicken or human caspase-1 and their cognate GSDM with the tetrapeptide as WT, YVAD, or AAAD. (**E**) Quantification of the intensity of the N-terminal band as a ratio of the N-terminal fragment over the full length, with the greatest ratio in a given experiment set to 100% efficiency. Each point is derived from a blot from a separate transfection. a, p < 0.05; b, p < 0.0001; c, p < 0.01 by one-way analysis of variance (ANOVA) with Tukey's multiple comparisons correction.

The online version of this article includes the following source data for figure 5:

**Source data 1.** AlphaFold prediction files in .pdb format.

**Source data 2.** Alignments used for visualizations.

**Source data 3.** Western blot densitometric analysis data.

**Source data 4.** Original file for the western blot analysis in *Figure 5D* (GSDMD, anti-FLAG).

**Source data 5.** PDF containing *Figure 5D* and original scans of the relevant western blot analysis (GSDMD, anti-FLAG) with highlighted bands and sample labels.

**Source data 6.** Original file for the western blot analysis in *Figure 5D* (GSDMA, anti-FLAG).

**Source data 7.** PDF containing *Figure 5D* and original scans of the relevant western blot analysis (GSDMA, anti-FLAG) with highlighted bands and sample labels.

This interaction uses hydrophobic resides found in the region between alpha helix α7' and α8 of the GSDMD C-terminus (**Liu et al., 2019**). We found this region is truncated in the AlphaFold prediction of chicken GSDMA (**Figure 5A, B**; **Table 1**; **Jumper et al., 2021**), preventing the interhelix loop from providing a groove for caspase-1. Furthermore, this region of chicken GSDMA does not contain

**Table 1.** Gasdermin sequences analyzed by AlphaFold.

| Species | GSDM | Accession |
|---|---|---|
| *Gallus gallus* | A | NP_001026532.1 |
| *Homo sapiens* | D | NP_001159709.1 |
| *Ornithorhynchus anatinus* | A | XP_039769487.1 |
| *Ornithorhynchus anatinus* | D | XP_028917767.1 |
| *Anolis carolinensis* | A | XP_008111551.1 |
| *Alligator mississippiensis* | A | XP_014451785.1 |
| *Gekko japonicus* | A | XP_015279010.1 |

hydrophobic residues (*Figure 5C*). This suggests that GSDMD inserted hydrophobic residues between the these α helices to better engage caspase-1. We hypothesized that chicken GSDMA instead fully relied on their tetrapeptide sequence for their activation by caspase-1.

Consistent with previous studies, human caspase-1 cleaves human GSDMD in a tetrapeptide-independent manner (*Figure 5D, E*). In contrast, activation of chicken GSDMA by chicken caspase-1 was lost when the FASD tetrapeptide was mutated to AAAD (*Figure 5D, E*). Because nearly a third of birds utilize a YVAD-like tetrapeptide in GSDMA, we modified the native chicken FASD to YVAD to determine if the precise sequence of the tetrapeptide affects cleavage efficiency. Indeed, YVAD was cleaved more efficiently than FASD (*Figure 5D, E*). We also mutated the human GSDMD tetrapeptide to YVAD but only observed slightly darker cleavage bands (*Figure 5D, E*). Therefore, the tetrapeptide sequence has a greater effect on cleavage efficiency for bird GSDMA compared to mammal GSDMD. These observations suggest that caspase-1 originally evolved alongside a gasdermin where cleavage was determined primarily by the tetrapeptide sequence. Because the YVAD-like tetrapeptides appear in a monophyletic group (*Figure 2B*), these data suggest that it was advantageous for some birds to use this sequence for GSDMA.

Together, these data suggest that GSDMA in birds does not use exosite-mediated interactions with caspase-1, and that these exosites developed after the gene duplication that gave rise to GSDMD, resulting in bird GSDMA fully relying on the tetrapeptide for cleavage.

## Reptile and amphibian GSDMA are cleaved by caspase-1

Birds share a close common ancestor with reptiles. To determine if bird GSDMA is representative of other non-mammal GSDMA, we evaluated the ability of GSDMA from *Alligator mississippiensis* (crocodilian reptile, American alligator) to be cleaved by the same caspase-1 proteins studied above. This reptile was selected as crocodilians share a more recent common ancestor with birds than with lizards and snakes (*Figure 2B*). Chicken, human, and mouse caspase-1 proteases could all cleave *A. mississippiensis* GSDMA (*Figure 6A*), which has a caspase-1 like FAHD tetrapeptide. The GSDMA tetrapeptides from more distantly related reptiles have more diverse tetrapeptide sequences. For example, GSDMA from *Anolis carolinensis* (reptile, Carolina anole) has a VVSD tetrapeptide. Even so, this GSDMA was also cleaved by all of these caspase-1 proteases (*Figure 6B*).

To more directly assess the origin of *GSDMA*, we looked to the clade that has the most ancient *GSDMA*, the amphibians (*Yuan et al., 2022*; *Wang et al., 2023b*). Although many amphibians have lost *GSDMA*, the caecilian order, which separate from frogs and salamanders early in the speciation of amphibians (*Upham et al., 2019*), including *Microcaecilia unicolor* (amphibian, tiny cayenne caecilian) retains it. All three tested caspase-1 proteins were all able to cleave this amphibian GSDMA (*Figure 6C*).

In further support of our hypothesis that GSDMA in all tetrapods except mammals serves as caspase-1 target gasdermin, in amphibians *GSDMA* is not enriched in the skin, but rather in the kidney (*Figure 6—figure supplement 1*; *Torres-Sánchez et al., 2019*), an organ that caecilians use for hematopoiesis (*Bain and Harr, 2022*), and also a site where *GSDMD* is enriched in mammals (*Uhlén et al., 2015*).

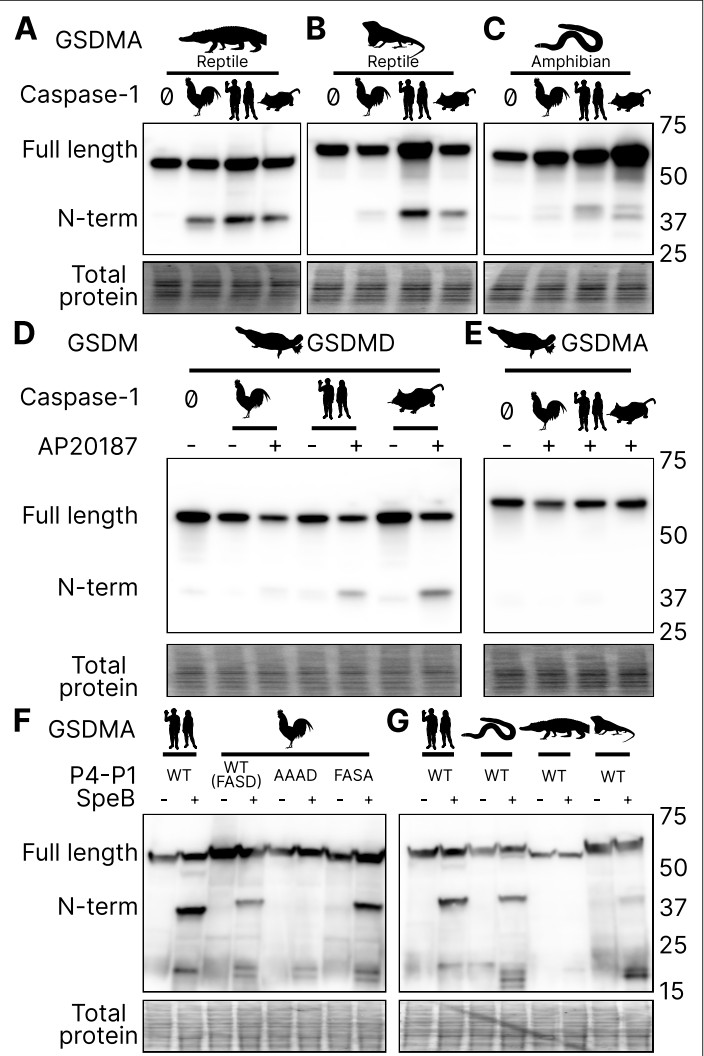

**Figure 6.** GSDMA from non-mammals are cleaved by caspase-1 and SpeB. (**A–C**) FLAG blot of HEK293T/17 lysates after co-transfection of *A. mississippiensis*, *A. carolinensis*, or *M. unicolor* GSDMA and caspase-1 from chicken, human, and mouse. *M. unicolor* is an amphibian despite sharing morphological similarity to a snake. (**D, E**) FLAG blot of HEK293T/17 lysates after co-transfection of platypus GSDMD or GSDMA with caspase-1 from chicken, human and mouse. (**F, G**) FLAG blot of HEK293T/17 lysates after transfection with plasmids encoding the designated gasdermins then incubation with SpeB. Total protein captured by Ponceau staining for this blot. Blots are representative of two biological replicates.

The online version of this article includes the following source data and figure supplement(s) for figure 6:

**Source data 1.** Original file for the western blot analysis in *Figure 6A, B* (anti-FLAG).

**Source data 2.** PDF containing *Figure 6A, B* and original scans of the relevant western blot analysis (anti-FLAG) with highlighted bands and sample labels.

**Source data 3.** Original file for the western blot analysis in *Figure 6C* (anti-FLAG).

**Source data 4.** PDF containing *Figure 6C* and original scans of the relevant western blot analysis (anti-FLAG) with highlighted bands and sample labels.

**Source data 5.** Original file for the western blot analysis in *Figure 6D* (anti-FLAG).

**Source data 6.** PDF containing *Figure 6D* and original scans of the relevant western blot analysis (anti-FLAG) with highlighted bands and sample labels.

**Source data 7.** Original file for the western blot analysis in *Figure 6E* (anti-FLAG).

**Source data 8.** PDF containing *Figure 6E* and original scans of the relevant western blot analysis (anti-FLAG) with highlighted bands and sample labels.

*Figure 6 continued on next page*

*Figure 6 continued*

**Source data 9.** Original file for the western blot analysis in *Figure 6F, G* (anti-FLAG).

**Source data 10.** PDF containing *Figure 6F, G* and original scans of the relevant western blot analysis (anti-FLAG) with highlighted bands and sample labels.

**Figure supplement 1.** Amphibian expression of *CASP1*, *GSDMA*, and *GSDME*.

## Platypus GSDMD but not GSDMA is cleaved by caspase-1

*O. anatinus* (monotreme mammal, platypus) GSDMD groups at the root node of mammal GSDMA proteins (*Figure 1*). The linker region of this gasdermin contains an FVKD tetrapeptide, but whether this divergent GSDMD was a target of caspase-1 had not been investigated. Again, all the caspase-1 proteases tested were able to cleave platypus GSDMD (*Figure 6D*), but not platypus GSDMA (*Figure 6E*). These data suggest that the ancestral *GSDMA* that gave rise to mammal *GSDMA* was indeed cleaved by caspase-1, and that this function was shifted into mammal GSDMD early in the evolution of mammals. In other words, caspase-1 cleaved GSDMA until GSDMD arose, allowing caspase-1 to free this ancient GSDMA to acquire a new function.

## SpeB broadly cleaves GSDMA proteins

Human and mouse GSDMA are cleaved by a bacterial cysteine protease called SpeB that is secreted by group A Streptococci during invasive skin infection. SpeB cleaves GSDMA and triggers pyroptosis in keratinocytes, where GSDMA is specifically expressed (*Deng et al., 2022*; *LaRock et al., 2022*; *Tamura et al., 2007*). Previously, this was the only known activator of any GSDMA. To determine if this cleavage is conserved in chicken GSDMA, we incubated lysates containing GSDMA with active SpeB and observed an N-terminal fragment that was consistent with cleavage in the GSDMA linker (*Figure 6F*). In humans, SpeB cleaves GSDMA in two nearby sites the linker region after VIL/IQ/ASD (*LaRock et al., 2022*; *Deng et al., 2022*). This protease preferentially cleaves peptides with a hydrophobic residue at P2 with negatively charged amino acids from P3' to P6'. SpeB demonstrates a preference for aromatic residues at P2 for some substrates (*Blöchl et al., 2021*), and indeed chicken GSDMA has a phenylalanine in the tetrapeptide.

We found that SpeB cleavage of chicken GSDMA was abrogated when using a AAAD tetrapeptide mutant (*Figure 6F*). Conversely, the negative charge from the P1 aspartic acid was dispensable (*Figure 6F*). These data reveal that the caspase-1 tetrapeptide of GSDMA in birds allows for cleavage by both caspase-1 and SpeB. We observed similar SpeB cleavage of GSDMA from *M. unicolor* and *A. carolinensis*, but not *A. mississippiensis* (*Figure 6G*). These data suggest that the ancestral GSDMA possessed the capacity to detect SpeB-like proteases within the cytosol of cells.

## Discussion

We discovered that GSDMA in amphibians, birds and reptiles are paralogs to mammal GSDMA. Surprisingly, the *GSDMA* genes in both the amphibians/reptiles/birds and mammal groups appear in the same locus. Therefore, this *GSDMA* gene was present in the common ancestor of all these animals. In mammals, this *GSDMA* duplicated to form *GSDMB* and *GSDMC*. Finally, a new gene duplicate, *GSDMD*, arose in a different chromosomal location. This *GSDMD* gene then became a superior target for caspase-1 after developing the exosite. Once *GSDMD* had evolved, we speculate that the mammalian *GSDMA* became a pseudogene that was available to evolve a new function. This new function included a new promoter to express mammalian *GSDMA* primarily in the skin, and perhaps acquisition of a new host protease that has yet to be discovered.

We also discovered that the ancestral state of GSDMA is to be cleaved by caspase-1. While bird GSDMA proteins share greater sequence homology with mammal GSDMA, bird GSDMA functionally behaves like mammal GSDMD. Caspase-1 is the activator of GSDMA not only in birds, but also in amphibians and reptiles, creating a continuous evolutionary thread where caspase-1 causes pyroptosis, though the target gasdermin has shifted after gasdermin gene duplication events. As our phylogenetic and activation data show, mammal GSDMA is the one GSDMA that deviates from all others. In future studies, the association of amphibian/bird/reptile GSDMA and caspase-1 should be confirmed in native cells from each of these animals.

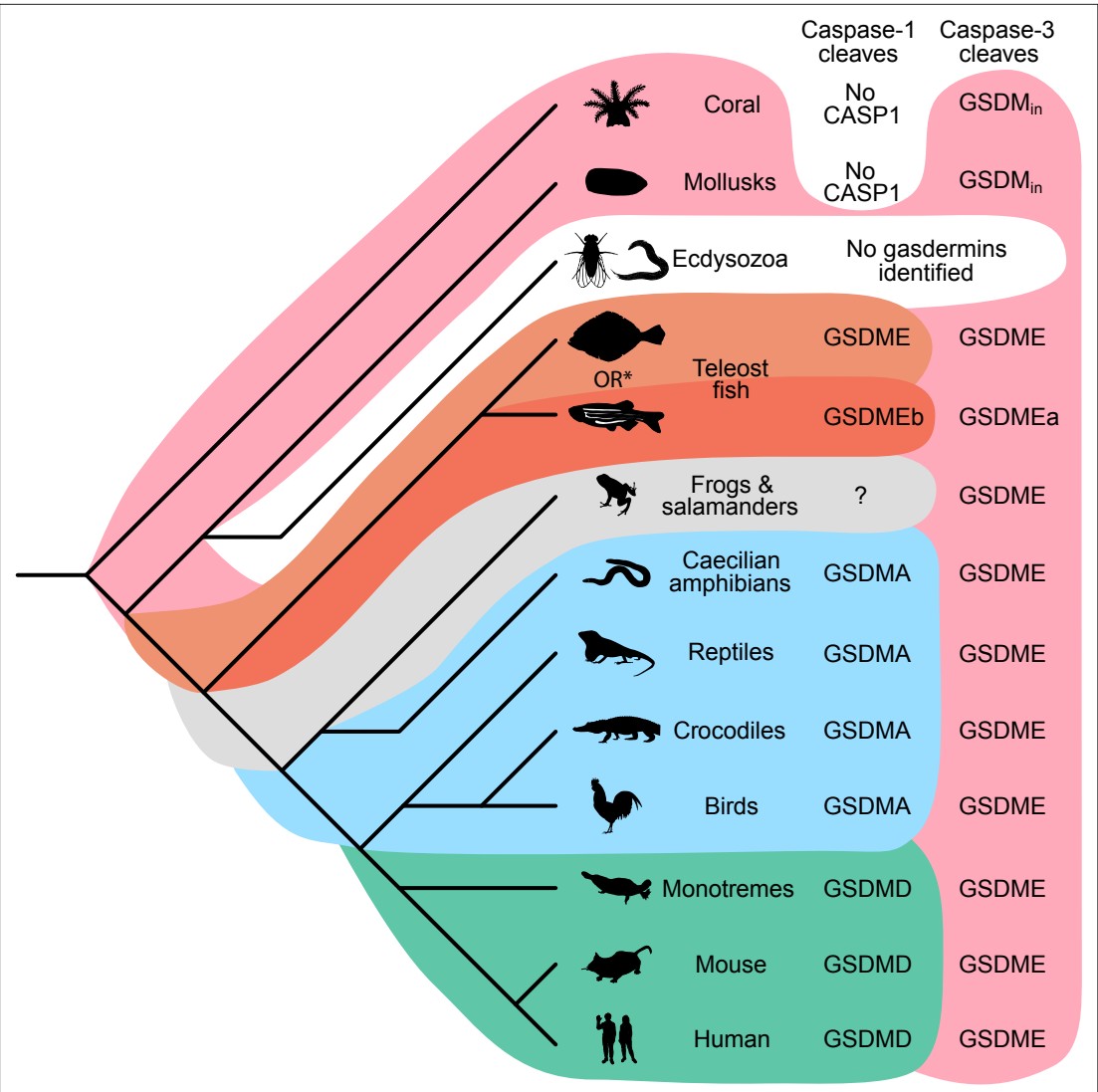

**Figure 7.** Caspase-1 targets different gasdermins throughout evolution. Phylogenetic tree indicating which GSDM caspase-1 cleaves in various clades of life. Branch lengths are not to scale. Coral, mollusks, hydras, and other invertebrates do not encode caspase-1 and cleave GSDM$_{in}$ with caspase-3. Coral and mollusks are indicated in this diagram. We did not identify any gasdermins in 3030 genomes from Ecdysozoa, the clade of which *D. melanogaster* and *C. elegans* are members. *Some teleost fish cleave GSDME with both caspase-1 and -3, however, other teleost fish, like the zebrafish, cleave GSDMEb with caspase-1 and GSDMEa with caspase-3, gasdermins that arose after a whole genome duplication event in teleost fish. Frogs and salamanders do not encode an identifiable *GSDMA*, but their GSDME contains prototypical caspase-3 cleavage sites. In mammals, GSDMD is cleaved by caspase-1 and GSDME by caspase-3. In caecilian amphibians—which are morphologically similar to snakes—birds, and reptiles, GSDMA is cleaved by caspase-1. We and others have confirmed that GSDME in birds is cleaved by caspase-3 (**Figure 3J**, **Figure 3—figure supplement 2**), and amphibian and reptile GSDME encodes prototypical caspase-3 cleavage sites.

Since mammal GSDMA is unusual, the open question is what new function that evolved for mammal GSDMA. Our tree provides a tantalizing clue: mammal GSDMA proteins are remarkably conserved compared to other gasdermins as seen in their branch lengths in our tree, suggesting that their function is similarly conserved, and may have an increased reliance on their precise sequence. This information may someday help reveal aspects of its activation.

Our data demonstrate that it is useful over evolutionary time to have several distinct ways to cause pyroptosis (**Figure 7**). GSDME in the phylogenetically oldest fish can be activated by either caspase-1 or -3 to cause pyroptosis. This leads to limited options in regulating pyroptosis, where expressing GSDME can convert caspase-3 signaling from an apoptotic outcome into a pyroptotic outcome. With the emergence of a second pore-forming gasdermin in amphibians (GSDMA) that could be activated by caspase-1 but resisted cleavage by caspase-3, these animals gained the ability to create new

dedicated caspase-1 signaling pathways that obligatorily cause pyroptosis. This development may enable more elaborate regulation of pyroptosis by enabling constitutive expression of inflammasome sensors, caspase-1, and GSDMA in immune cells without fear of errantly converting apoptosis to pyroptosis. This separation of caspase activation mechanisms is ensured both by the different tetrapeptide specificities of caspase-1 and -3, and by caspase-3 inactivating GSDMA. Separation of caspase-1- and caspase-3-dependent pyroptosis must be of great utility, as the exact same adaptation independently evolved in some teleost fish (*Christoffels et al., 2004*; *Hoegg et al., 2004*; *Meyer and Van de Peer, 2005*), where GSDMEa is cleaved by caspase-3, and GSDMEb is cleaved by caspase-1.

Throughout most of evolutionary history, these two mechanisms were sufficient to provide specificity. With the expansion of gasdermins in mammals, the complexity of ensuring specificity for gasdermin activation increased. Mammals need to separate five different gasdermin activating pathways while simultaneously maintaining the efficiency of each pathway. We speculate that this could provide an evolutionary driving force that caused mammals to make significant structural changes that pair caspase-1 more specifically with GSDMD using exosite interactions between the two. The exosite could enhance existing properties of the gasdermin, for example enhancing cleavage by caspase-1 (and caspase-4/5/11) or minimizing cleavage by caspase-3. Alternately, the exosite could prevent GSDMD from being cleaved by the new activating pathways for mammal GSDMA, GSDMB, and GSDMC. Another possibility is that the exosite may have permitted the GSDMD to be cleaved by caspase-8 (*Orning et al., 2018*; *Sarhan et al., 2018*), a caspase that typically triggers apoptosis that could not change its tetrapeptide specificity. Overall, we speculate that after the duplication event, the new gene that gave rise to *GSDMD* developed the exosites, and that this was superior to the parental *GSDMA*, which then allowed mammal *GSDMA* to evolve new functions no longer tied to caspase-1.

Sequestering GSDMA, B, C, and D from apoptotic caspases makes intuitive sense. The expansion of the gasdermin family in mammals enabled hosts to fine tune the activation of pyroptosis in ways that necessitated distinct—often cell extrinsic—activation pathways and biochemical properties. For example, GSDME is targeted more specifically to mitochondria in neurons (*Neel et al., 2023*). Additionally, GSDMD and GSDMA control the release of cellular contents based on their charge (*Xia et al., 2021*). Some gasdermins may push cells to pyroptosis faster than others, allowing cells more or less time to complete tasks before the cell lyses. In mammals, activating GSDMD in intestinal epithelial cells causes extrusion that must be completed before the cell lyses (*Nozaki et al., 2022b*). Activating the wrong pyroptotic pathway may not allow a cell adequate time to complete these 'bucket list' tasks that promotes appropriate disposal of dying cells (*Nozaki et al., 2022a*).

The gasdermin family is quickly evolving. Nearly doubling the number of gasdermins provided mammals an opportunity to attach diverse signaling pathways as initiators of pyroptotic cell death, and finely tune their activation. We speculate that seizing this opportunity was a key step in the evolution of the mammalian immune system.

**Table 2.** Two letter species abbreviations used in this study.

| Abbreviation | Full name | Common name |
|---|---|---|
| Hs | *Homo sapiens* | Human |
| Gg | *Gallus gallus* | Chicken |
| Am | *Alligator mississippiensis* | American alligator |
| Ac | *Anolis carolinensis* | Carolina anole |
| Mu | *Microcaecilia unicolor* | Tiny cayenne caecilian |
| Oa | *Ornithorhynchus anatinus* | Platypus |
| Mm | *Mus musculus* | Mouse |

## Materials and methods

### Phylogenetic dataset construction

Putative gasdermins proteins were obtained using three complementary approaches: keyword search on the NCBI website, protein similarity searches in NCBI nr (https://www.ncbi.nlm.nih.gov/; last accessed 2021), and UniProt (*Bateman et al., 2021*) (last accessed 2021) using human gasdermins sequences as seed using BLASTp (*Altschul et al., 1997*) (e-value of 0.01 as cut-off). Retrieved sequenced were combined with gasdermins sequences used by *De Schutter et al., 2021*. We partitioned sequences according to their type (e.g., A, B, C, and D). Redundant proteins were removed by clustering using CD-HIT (*Li and Godzik, 2006*). We keep only one sequence per sequence (1-to-1 orthologs) using a reciprocal best BLASTp approach (e-value $<10^{-5}$ as threshold). For each set, multiple protein alignments were generated using Clustal Omega (*Sievers et al., 2011*). Partial sequences were removed using trimAl (*Capella-Gutiérrez et al., 2009*) (criteria <50% of alignment length, >50% Gap) and/or visual inspection in Jalview (*Waterhouse et al., 2009*). Curated alignments were used to produce for Hidden Markov models and consensus sequences using HMMER (http://hmmer.org/), and protein profiles for AUGUSTUS-PPX (*Keller et al., 2011*). Abbreviations for animal genera and species are listed in *Table 2*.

### Genome annotation

Genome sequences for selected species were downloaded from NCBI. Corresponding raw reads (Illumina fastq files) were downloaded from NCBI SRA when available. For species with annotated protein genes, we searched all gasdermins using BLASTp and if absent we used TBLASTn with an e-value cut-off of 0.1 against the genome (*Altschul et al., 1997*). If no hit found, a local de novo assembly from raw reads using Spades (*Bankevich et al., 2012*) was attempted. This has had a limited success because many genomes were constructed using older versions of Illumina reads (50 nucleotides).

### Screening of genomes to identify gasdermin sequences

The purpose of the initial screening was to flag genomic regions with putative gasdermins for further annotation. All genomes were initially screened with exonerate (*Slater and Birney, 2005*) using consensus sequences as seed with relaxed parameters (see GitHub, copy archived at *Cisse, 2023*). Protein to genome alignments were filtered for spurious alignments and converted in transcripts using gffread utility (*Pertea and Pertea, 2020*), and protein using Funannotate utility 'gff2prot' utility (*Palmer and Stajich, 2022*).

Once validated, all gasdermin putative loci were annotated using AUGUSTUS-PPX (*Keller et al., 2011*) and/or BRAKER (*Brůna et al., 2021*). For species with RNA-seq data were available, RNA-seq reads were mapped to the corresponding genomes using hisat2 (*Kim et al., 2019*). Alignments were converted in BAM formats using SAMTOOLS (*Li et al., 2009*) to generate BAM alignments for AUGUSTUS prediction (BAM2hints, *Stanke et al., 2006*).

### Phylogenetic placement

Predicted gasdermins were aligned using MAFFT (*Katoh et al., 2002*). The resulting alignment was inspected for inconsistencies in Jalview (*Waterhouse et al., 2009*). Maximum likelihood phylogenies were inferred using FastTree (*Price et al., 2009*) and raxml-ng (*Kozlov et al., 2019*). Time-calibrated trees were constructed using BEAST (*Bouckaert et al., 2014*) with log-normal distribution and strict clock using soft calibrations from mammal fossil estimated ages. Species age calibrations from https://vertlife.org/ (*Upham et al., 2019*).

### Cloning and mutagenesis

Chicken *GSDMA*, *GSDME*, and *CASP1* were isolated from cDNA from chicken tissue. All animals used for this purpose were treated in accordance to Chicken caspase-1 was previously reported to express a short 283 amino acid protein that is truncated when compared to human caspase-1 (*Johnson et al., 1998*). We cloned a full-length, 393 amino acid isoform from chicken spleens based on more recent NCBI sequences. CARD domains were removed as determined by NCBI annotations. Chicken, crow, mouse, and human GSDM were cloned into pcDNA3 with N-3xFLAG tag. Site-based mutagenesis was performed using in vivo assembly (*Watson and García-Nafría, 2019*; *García-Nafría et al., 2016*).

**Table 3.** Selected primers used in this study.

| Primer | Sequence |
| --- | --- |
| GgA-F | CTCAGACCAAGACCACTTCAGGC |
| GgA-R | CTGGGAAGGAGGGACTCAACGC |
| GgE-F | CTCACCTCCTCATCAGTGCTATCT |
| GgE-R | TGAAGCGATGTAAGGCAAGCAAC |
| GgCasp1-F | CGAGTACCAGCGCATCAAGGA |
| GgCasp1-R | TGACTGTGGTCATCTCCTGGGAA |
| Gg18S-F | TGTGCCGCTAGAGGTGAAATT |
| Gg18S-R | TGGCAAATGCTTTCGCTTT |

Caecilian, alligator, anole, and platypus gasdermins were synthesized by TwistBioscience (South San Francisco, CA, USA) and cloned using Gateway technology (Thermo Fisher, Waltham, MA, USA) into a pCMV-polysite (*Agrotis et al., 2019*) plasmid modified to include N-terminal 3xFLAG tag using the primers listed.

## qRT-PCR analysis of chicken tissues

The tissue samples were stored in RNAlater RNA Stabilization Solution (AM7020, Thermo Fisher, Waltham, MA, USA) at −80°C if the experiments were not processed immediately. Approximately 20 µg tissue samples were harvested for RNA extraction with TRIzol reagent (15596026, Thermo Fisher, Waltham, MA, USA), and 1 µg total RNA was used for reverse transcription through Super-Script IV Reverse Transcriptase (18090200, Thermo Fisher, Waltham, MA, USA). The cDNA samples were diluted 80 times with molecular biology grade water for the PCR experiments. The genes were tested by normal SYBR green method with PowerTrack SYBR Green Master Mix (A46110, Thermo Fisher, Waltham, MA, USA). qRT-PCR results were analyzed by ΔΔCT method. Primers used are listed in *Table 3*.

## Cell culture maintenance

HEK293T/17 (CRL-11268, ATCC, VA, USA) and HeLa (CCL-2, ATCC, VA, USA) cell lines were maintained in Dulbecco's modified Eagle medium (DMEM; 11995-073, Thermo Fisher, Waltham, MA, USA) supplemented with 1% minimal essential medium non-essential amino acids solution (11140-050, Thermo Fisher, Waltham, MA, USA) and 10% fetal bovine serum (SH30396.03, Cytiva, Utah, USA). Cell lines were purchased from the Duke Cell Culture facility where they underwent STR authentication. Cell lines were tested regularly for mycoplasma infection by PCR (*Dreolini et al., 2020*). Cell lines were kept under 25 passages after thawing.

## Transfection and dimerization

HEK293T/17 cells were transfected using Lipofectamine 3000 (L3000008, Thermo Fisher, Waltham, MA, USA) according to the user manual. All cells received the same amount of DNA with backbone pcDNA3 making up differences in mass. AP20187 (HY-13992, MedChemExpress, Monmouth Junction, NJ, USA) was added to cells by replacing supernatant with OptiMEM (31985-070, Thermo Fisher, Waltham, MA, USA) with 500 nM AP20187 or DMSO for western blot endpoints, or was added to the supernatant for a final concentration of 500 nM in complete DMEM for LDH endpoints. Plasmids used are listed in *Table 4*.

## Western blotting

HEK293T/17 cells were lysed directly in 1xLSB, boiled for 5 min then ran on SDS–PAGE using Bio-Rad TGX gels (Hercules, CA, USA). Total protein was captured after activation with UV for 5 min on Azure Biosystems Azure 500 imager (Dublin, CA, USA). Protein was transferred to membrane using a Bio-Rad Trans-Blot Turbo transfer system. Total protein images used in figures were captured on membranes

**Table 4.** Plasmids used in this study.

| Plasmid ID | Source |
| --- | --- |
| pHom-Mem1 | Takara 635064 |
| pMSCV-F-del Casp9.IRES.GFP | Addgene 15567 |
| pDEST-CMV-polysite | Addgene 122843 |
| pcDNA3.1(+) | Thermo Fisher V790-20 |
| pEGFP-N1 | U55762.1 |
| pmCherry-N1 | Takara 632523 |
| 2xDmrB-Δ CARD-GgCASP1 | This study |
| 2xDmrB-Δ CARD-HsCASP1 | This study |
| 2xDmrB-Δ CARD-MmCASP1 | This study |
| 3xFLAG-N-GgGSDMA | This study |
| 3xFLAG-N-GgGSDMA-D244A | This study |
| 3xFLAG-N-GgGSDMA-AAAD | This study |
| NoTag-GgGSDMA | This study |
| NoTag-GgGSDMA-D244A | This study |
| 3xFLAG-N-GgGSDME | This study |
| 3xFLAG-N-GgGSDME-D275A | This study |
| 3xFLAG-N-HsGSDMA | This study |
| 3xFLAG-N-HsGSDMD | This study |
| 3xFLAG-N-HsGSDMD-D275A | This study |
| 3xFLAG-N-HsGSDMD-AAAD | This study |
| 3xFLAG-N-MmGSDMD | This study |
| 3xFLAG-N-MmGSDMD-D276A | This study |
| 3xFLAG-N-MuGSDMA | This study |
| 3xFLAG-N-AmGSDMA | This study |
| 3xFLAG-N-AcGSDMA | This study |
| 3xFLAG-N-OaGSDMA | This study |
| 3xFLAG-N-OaGSDMD | This study |

by UV after transfer unless otherwise specified. For densitometric analysis of bands, ImageJ was used (*Schindelin et al., 2012*). Antibodies used in western blotting are listed in *Table 5*.

## N-terminal lysis assay

HEK293T/17 cells were transfected with EGFP and either the full length or N-terminal domain of chicken GSDMA without a 3xFLAG tag. After 48 hr, GFP fluorescence was captured on a BioTek Instruments Synergy H1 microplate reader (Winooski, VT, USA). After all experiments were captured, the GFP full-length intensity was normalized to 100%. N-terminal lysis data were transformed linearly by the same scale as their accompanying full-length data.

**Table 5.** Antibodies used for western blotting in this study.

| Antibody | Cat # | Manufacturer |
| --- | --- | --- |
| M2 FLAG | F1804 | Sigma-Aldrich, Burlington, MA, USA |
| Goat-anti-mouse HRP | 115-035-062 | Jackson ImmunoResearch, West Grove, PA, USA |

## LDH cell death assay

HEK293T/17 cells were transfected with chicken caspase-1 and either wild type or D244A mutant GSDMA without a 3xFLAG tag. After 48 hr, AP20187 dimerizer was added in cell culture media to a final concentration of 500 nM for 5 hr before taking the supernatants for LDH assay using CytoTox 96 Non-Radioactive Cytotoxicity Assay (G1780, Promega, Madison, WI). Cell death was measured on a scale with the media without cells set to 0% and cells lysed for 1 hr in 1% Triton X-100 set to 100%.

## Microscopy

HeLa cells were plated on Lab-Tek II chambered coverglass (155360, Thermo Fisher, Waltham, MA, USA). Cells were transfected with either chicken N-terminal GSDMA domain or equivalent mass of pmCherry. After 18 hr, dyes were added to a final concentration of 0.5 ng/ml Hoechst (H3570, Thermo Fisher, Waltham, MA, USA) and 160 nM SYTOX Green (S7020, Thermo Fisher, Waltham, MA, USA) to each well. After 30-min incubation cells were imaged at ×20 and ×60 on a Keyence BZ-X810 microscope (Osaka, Japan).

## AlphaFold prediction

C-terminal domains from the following proteins were visualized using the ColabFold notebook (*Mirdita et al., 2022*; *Mitchell et al., 2019*; *Mirdita et al., 2017*; *Berman et al., 2003*; *van Kempen et al., 2024*; *Steinegger et al., 2019*; *Mirdita et al., 2019*). C-terminal domains processed included all the protein after the predicted aspartic acid cleavage site in the linker region.

## Pymol visualization

Pymol (*Schrödinger, 2015*) version 2.6.0a0 (26d795fe92) was compiled from source.

## SpeB cleavage assays

SpeB (20 Units, ACRO biosystems, SPB-S5115, Newark, DE, USA) was incubated with cell lysates of 293T transfected with FLAG-tagged gasdermin A for 30 min at 37°C. For transient transfection for 293T/17, followed manufacturer's instructions of Lipofectamine 2000 (Thermo Fisher, 11668019, Waltham, MA, USA). Cell lysates were run on sodium dodecyl sulfate–polyacrylamide gel electrophoresis (SDS–PAGE) and transferred to nitrocellulose membrane. Immunoblotted using M2 anti-FLAG antibody. For these experiments, total protein was detected by Ponceau S staining (Thermo Fisher, A40000279, Waltham, MA, USA).

## R statistical analysis and visualization

R version 4.3.0 was used in these studies. Plots were generated using ggplot2 version 3.4.2.

## Animal silhouettes

All animal silhouettes are from PhyloPic and are free of known restrictions or are dedicated to the public domain under the following licenses.

| Animal | Creator | License |
|---|---|---|
| Human | Yan Wong | Public Domain Mark 1.0 |
| Mouse | T. Michael Keesey | Public Domain Mark 1.0 |
| Platypus | Steven Traver | CC0 1.0 Universal Public Domain Dedication |
| Caecilian | Chuanxin Yu | CC0 1.0 Universal Public Domain Dedication |
| Chicken | Steven Traver | CC0 1.0 Universal Public Domain Dedication |
| Crow | T. Michael Keesey | CC0 1.0 Universal Public Domain Dedication |
| Alligator | Ferran Sayol | CC0 1.0 Universal Public Domain Dedication |
| Anole | Ingo Braasch | CC0 1.0 Universal Public Domain Dedication |
| Shark | Ingo Braasch | CC0 1.0 Universal Public Domain Dedication |

*Continued on next page*

*Continued*

| Animal | Creator | License |
|--------|---------|---------|
| Coral | Christoph Schomburg | CC0 1.0 Universal Public Domain Dedication |
| Mollusk | Riccardo Percudani | CC0 1.0 Universal Public Domain Dedication |
| Fly | Andy Wilson | CC0 1.0 Universal Public Domain Dedication |
| Nematode | Jake Warner | CC0 1.0 Universal Public Domain Dedication |
| Flatfish | Nathan Hermann | CC0 1.0 Universal Public Domain Dedication |
| Zebrafish | Jake Warner | CC0 1.0 Universal Public Domain Dedication |
| Frog | Steven Traver | CC0 1.0 Universal Public Domain Dedication |

## Acknowledgements

We would like to thank Dr. Kulkarni from NCSU for graciously providing chicken tissues for qRT-PCR analysis. We would like to thank Dr. Luke Borst for the generous gift of chicken tissues used for isolating the chicken GSDMA, GSDME, and CASP1 sequences. We would like to thank Dr. Jörn Coers for the HeLa cells used in this study. This work has been funded in whole or in part with federal funds from the Intramural Research Program of the US National Institutes of Health (NIH) Clinical Center. The views expressed in this work are those of the authors and do not necessarily represent the official positions of the NIH or U.S. Government. This study used the Office of Cyber Infrastructure and Computational Biology (OCICB) High Performance Computing (HPC) cluster at the National Institute of Allergy and Infectious Diseases (NIAID), Bethesda, MD. This study also utilized the high-performance computational capabilities of the Biowulf Linux cluster at the NIH, Bethesda, MD (http://biowulf.nih.gov). pDEST-CMV-polysite was a gift from Robin Ketteler (Addgene plasmid # 122843; https://www.addgene.org/122843/; RRID:Addgene_122843). pMSCV-F-del Casp9.IRES.GFP was a gift from David Spencer (Addgene plasmid # 15567; https://www.addgene.org/15567/; RRID:Addgene_15567). Silhouettes in figures are from PhyloPic. All of these cartoons have been donated to the public domain. This work was not submitted to other journals before eLife.

## Additional information

### Funding

| Funder | Grant reference number | Author |
|--------|------------------------|--------|
| National Institutes of Health | AI133236 | Edward A Miao |
| National Institutes of Health | AI139304 | Edward A Miao |
| National Institutes of Health | AI136920 | Edward A Miao |
| National Institutes of Health | AI148302 | Edward A Miao |
| National Institutes of Health | AI175078 | Edward A Miao |
| National Institutes of Health | AR072694 | Edward A Miao |

The funders had no role in study design, data collection, and interpretation, or the decision to submit the work for publication.

## Author contributions
Zachary Paul Billman, Conceptualization, Data curation, Formal analysis, Validation, Investigation, Visualization, Methodology, Writing – original draft, Writing – review and editing; Stephen Bela Kovacs, Conceptualization, Data curation, Investigation, Methodology, Writing – review and editing; Bo Wei, Kidong Kang, Investigation; Ousmane H Cissé, Resources, Data curation, Software, Formal analysis, Validation, Investigation, Methodology, Writing – review and editing; Edward A Miao, Funding acquisition, Project administration, Writing – review and editing

## Author ORCIDs
Zachary Paul Billman ⓘ https://orcid.org/0000-0001-7738-5795
Ousmane H Cissé ⓘ https://orcid.org/0000-0002-2990-2185
Edward A Miao ⓘ http://orcid.org/0000-0001-7295-3490

Reviewer #1 (Public Review): https://doi.org/10.7554/eLife.92362.4.sa1
Reviewer #2 (Public Review): https://doi.org/10.7554/eLife.92362.4.sa2
Author Response https://doi.org/10.7554/eLife.92362.4.sa3

# Additional files

## Supplementary files
• MDAR checklist

## Data availability
All phylogenetic analyses and detailed pipeline are open source and can be found at https://github.com/ocisse/gasdermin, (copy archived at *Cisse, 2023*). All plasmids created for this study are available upon request. Source data for all figures, including numerical data for graphs, uncropped western blots, and sequences used to build phylogeny are included in this submission.

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
