## [Editor Report · eLife assessment]

This study presents **valuable** insights into the evolution of the gasdermin family, making a strong case that a GSDMA-like gasdermin activated by caspase-1 cleavage was already present in early land vertebrates. **Convincing** biochemical evidence is provided that extant avian, reptilian, and amphibian GSDMA proteins can still be activated by caspase-1 and upon cleavage induce pyroptosis-like cell death -- at least that they do so in the context of human cell lines. The caspase-1 cleavage site has only been lost in mammals, which use the more recently evolved GSDMD as a caspase-1 cleavable pyroptosis inducer. The presented work will be of considerable interest to scientists working on the evolution of cell death pathways, or on cell death regulation in non-mammalian vertebrates.

---

## [Referee Report · Reviewer #1 (Public Review)]

Summary:

In their revised manuscript, the authors analyze the evolution of the gasdermin family and observe that the GSDMA proteins from birds, reptiles and amphibians does not form a clade with the mammalian GSDMAs. Moreover, the non-mammalian GSDMA proteins share a conserved caspase-1 cleavage motif at the predicted activation site. The authors provide several series of experiments showing that the non-mammalian GSDMA proteins can indeed be activated by caspase-1 and that this activation leads to cell death (in human cells). They also investigate the role of the caspase-1 recognition tetrapeptide for cleavage by caspase-1 and for the pathogen-derived protease SpeB.

Strengths:

The evolutionary analysis performed in this manuscript appears to use a broader data basis than what has been used in other published work. An interesting result of this analysis is the suggestion that GSDMA is evolutionary older than the main mammalian pyroptotic GSDMD, and that birds, reptiles and amphibians lack GSDMD but use GSDMA for the same purpose. The consequence that bird GSDMA should be activated by an inflammatory caspase (=caspase1) is convincingly supported by the experiments provided in the manuscript.

The changes made by the authors in response to the previous reviewer comments are (in my opinion) sufficient.

---

## [Referee Report · Reviewer #2 (Public Review)]

Summary:

The authors investigated the molecular evolution of members of the gasdermin (GSDM) family. By adding the evolutionary time axis of animals, they created a new molecular phylogenetic tree different from previous ones. The analyzed result verified that non-mammalian GSDMAs and mammalian GSDMAs have diverged into completely different and separate clades. Furthermore, by biochemical analyses, the authors demonstrated non-mammalian GSDMA proteins are cleaved by the host-encoded caspase-1. They also showed mammalian GSDMAs have lost the cleavage site recognized by caspase-1. Instead, the authors proposed that the newly appeared GSDMD is now cleaved by caspase-1.

Through this study, we have been able to understand the changes in the molecular evolution of GSDMs, and by presenting the cleavage of GSDMAs through biochemical experiments, we have become able to grasp the comprehensive picture of this family molecules. However, there are some parts where explanations are insufficient, so supplementary explanations and experiments seem to be necessary.

Strengths:

It has a strong impact in advancing ideas into the study of pyroptotic cell death and even inflammatory responses involving caspase-1.

Weaknesses:

Based on the position of mammalian GSDMA shown in the molecular phylogenetic tree (Figure 1), it may be difficult to completely agree with the authors' explanation of the evolution of GSDMA.

(1) Focusing on mammalian GSDMA, this group and mammalian GSDMD diverged into two clades, and before that, GSDMA/D groups and mammalian GSDMC separated into two, more before that, GSDMB, and further before that, non-mammalian GSDMA, when we checked Figure 1. In the molecular phylogenetic tree, it is impossible that GSDMA appears during evolution again. Mammalian GSDMAs are clearly paralogous molecules to non-mammalian GSDMAs in the figure. If they are bona fide orthologous, the mammalian GSDMA group should show a sub-clade in the non-mammalian GSDMA clade. It is better to describe the plausibility of the divergence in the molecular evolution of mammalian GSDMA in the Discussion section.

(2) Regarding (1), it is recommended that the authors reconsider the validity of estimates of divergence dates by focusing on mammalian species divergence. Because the validity of this estimation requires recheck of the molecular phylogenetic tree, including alignment.

(3) If GSDMB and/or GSDMC between non-mammalian GSDMA and mammalian GSDMD as shown in the molecular phylogenetic tree would be cleaved by caspase-1, the story of this study becomes clearer. The authors should try that possibility.

---

## [Author Response]

The following is the authors’ response to the previous reviews.

**Recommendations for the authors:**

**Reviewer #1 (Recommendations For The Authors):**
The authors have addressed the specific comments made upon the initial submission. In particular, they have now provided an explanation, why their GSDM tree looks different than previously published trees. The authors have also followed my initial suggestion to consider the highly-conserved residue following the cleavage site in bird GSDMA forms. Some of the more general weaknesses remain, since they cannot easily be addressed. I agree with the suggestions made by reviewer #2 to further improve the manuscript.

We thank the reviewer for their insight which we think has improved our manuscript. We have additionally made the changes requested by this reviewer and reviewer #2 in the next section.

**Reviewer #2 (Recommendations For The Authors):**
The authors responded sincerely to our reviewers' questions in the revised manuscript and I sufficiently understand. After re-reading it, however, I found two issues that need to be revised, so please consider doing them.(1) New sentences (Page 5, lines 209-212) that the authors have added are better written in the subsection, "Bird GSDMA is activated .." after some modification. Because there is an undeniable sense of suddenness in present position.

We agree with this evaluation and have moved these sentences to a more natural position in the following section.

(2) Regarding the chromosomal location of the GSDMA gene, the authors describe that the genes of mammals, birds, and reptiles localize the same genetic locus, but no data are presented. To support their claim, it should also be presented as a supplementary figure.

We agree with this evaluation and have generated Figure 1 – Supplemental 4 to show the synteny of the GSDMA locus from humans to GSDMEc in sharks.